# Efficacy of traditional Chinese medicine injections for treating idiopathic pulmonary fibrosis: A systematic review and network meta-analysis

Shuai-yang Huang[1,2], Hong-sheng Cui [2]*, Ming-sheng Lyu[1,2,3], Gui-rui Huang[1,2], Dan Hou[1,2], Ming-xia Yu[1,2]

1 Beijing University of Chinese Medicine, Beijing, China, 2 The Third Affiliated Hospital of Beijing University of Chinese Medicine, Beijing, China, 3 Beijing University of Chinese Medicine Affiliated Dongzhimen Hospital, Beijing, China

* Hshcui@sina.com

**Data Availability Statement:** All relevant data are within the article and its Supporting information files.

## Abstract

### Background

Idiopathic pulmonary fibrosis (IPF), acutely or slowly progressing into irreversible pulmonary disease, causes severe damage to patients' lung functions, as well as death. In China, Chinese medicine injections (CMIs) have been generally combined with Western medicine (WM) to treat IPF, which are safe and effective. This study aimed to systematically compare the efficacy of 14 CMIs combined with WM in the treatment of IPF based on a systematic review and network meta-analysis (NMA).

### Material and methods

PubMed, Web of Science, Embase, Cochrane Library, MEDLINE, and Chinese databases, including the China National Knowledge Infrastructure, Wanfang Database, Scientific Journal Database, and China Biology Medicine Database were searched from inception to October 31, 2021. The inclusion criterion was randomized controlled trials (RCTs) on CMIs with WM for treating IPF. Reviewers independently screened the literature, extracted data, and evaluated the risk of bias in the included studies. RevMan 5.4 software and Stata software (version 16.0) were used for the data analysis. NMA were carried out for calculating the odd ratios (ORs) with 95% confidence intervals (CI), the surface under cumulative ranking curve (SUCRA) and the probabilities of being the best.

### Results

A total of 63 eligible RCTs involving 14 CMIs were included in this NMA. More CMIs can significantly improve the clinical effectiveness rate (CER); Shuxuening injection (SXN)+WM (OR 8.91, 95% CI 3.81–20.83), Shuxuetong injection (SXT)+WM (OR 7.36, 95% CI 3.30–16.00), Shenxiong injection (SX)+WM (OR 5.42, 95% CI 2.90–10.13), Danhong injection (DH)+WM (OR 4.06, 95% CI 2.62–6.29), and Huangqi injection (HQ)+WM (OR 3.47, 95% CI 1.55–7.77) were the top five treatment strategies. Furthermore, DH +WM ranked

**Funding:** The author(s) received no specific funding for this work.

**Competing interests:** The authors have declared that no competing interests exist.

relatively high in the SUCRA value of the nine outcome indicators, oxygen partial pressure (PaO2) (OR -13.39; 95% CI -14.90,-11.89; SUCRA 83.7%), carbon dioxide partial pressure (PaCO2) (OR -4.77; 95% CI -5.55,-3.99; SUCRA 83.3), orced vital capacity (FVC) (OR -1.42; 95% CI -2.47,-0.36; SUCRA 73.5%), total lung capacity (TLC) (OR 0.93; 95% CI 0.51,1.36; SUCRA 89.0%), forced expiratory volume 1/ forced vital capacity (FEV1/FVC%) (OR -10.30; 95% CI -12.98,-7.62; SUCRA 72.7%), type III collagen (IIIC) (OR 13.08; 95% CI 5.11,21.05; SUCRA 54.9%), and transforming growth factor (TGF) (OR -4.22; 95% CI -6.06,-2.37; SUCRA 85.7%) respectively, which seems to indicate that DH+WM had the highest likelihood of being the best treatment.

## Conclusions

This review specified several CMIs combined with WM in the treatment of IPF in China. In contrast to glucocorticoids or antioxidants, CMIs combined with WM delayed the decline in lung function, maintained oxygenation and quality of life in patients with IPF. The combined use of DH, SXN, SX, and safflower yellow sodium chloride injection (HHS) with WM exerted a more positive effect in treating IPF than WM alone. However, there were limitations to the conclusions of this study due to quality control differences in the included trials.

## 1 Introduction

Idiopathic pulmonary fibrosis (IPF) is a chronic, progressive and lethal fibrotic lung disease, characterized by diffuse alveolitis, profound changes in epithelial cell phenotype and fibroblast proliferation. The incidence of IPF is around 8/10 million-15/10 million, accounting 65% interstitial lung disease [1]. IPF mostly presents as a chronic disease, but patients have an average median survival of only 2–4 years after diagnosis. Because of its unclear pathogenesis, the treatments for IPF are limited and causing high rate of mortality [2].

Glucocorticoids can relieve IPF patients' symptoms, but it is ineffective in reversing the lung damage. Lung transplantation is the last treatment for IPF patients. More effective therapeutic ways are becoming available for IPF patients following the research progress on pathogenesis of IPF [3]. Pirfenidone and nidanib are approved for the treatment of IPF because they can slow down the decline of lung function and disease progression; however, these two drugs have more adverse effects, and no reliable evidence has been found to confirm that they significantly improve patients' symptoms and quality of life. Moreover, the cost of both drugs is high, which places a heavy economic burden on patients and society [4]. Traditional Chinese medicine (TCM) has a long history of treating IPF, and treatment is based on syndrome differentiation, which has the advantages of low toxicity, multi-level and multi-target, and unique advantages in clinical application [5, 6].

Clinical trials of TCM in the treatment of IPF are gradually increasing, but it is unclear whether they can slow down the progression of the disease. Based on the research method of evidence-based medicine, this study used NMA to compare the number of different CMIs combined with WM interventions under the same conditions to obtain more reliable evidence for clinical reference. NMA is a development of the traditional meta-analysis, which has the advantage of providing quantitative statistical analysis of different interventions for the same disease and ranking them in order of efficacy, thus providing evidence to support the clinical use of drugs. Therefore, This study aims to provide evidence-based clinical practice by collecting RCTs on the efficacy and safety of CMIs combined with WM in the treatment of IPF.

## 2 Methods

The protocol of this study was registered and approved by the International Prospective Register of Systematic Reviews (PROSPERO) on January 19, 2022 (registration number CRD42022295916). This study was conducted in strict accordance with this protocol. This study has no ethical implications.

### 2.1 Search strategy

A comprehensive search of PubMed, Web of Science, Embase, The Cochrane Library, MED-LINE, CNKI, Wanfang database, VIP, and China CBM was performed by two researchers. The search terms included Pulmonary Fibroses, Fibroses, Pulmonary, Fibrosis, Pulmonary, Alveolitis, Fibrosing, Alveolitides, Fibrosing, Fibrosing Alveolitides, Danhong, and Danhong injection. Detailed search strategies are described in S3 File.

### 2.2 Inclusion criteria

The inclusion criteria were as follows: (I) Study design: only RCTs were included; (II) Participants: patients diagnosed with IPF. For the sake of comprehensiveness of the evidence, we placed no restrictions on the severity of the disease and other features, such as typology or complications; (III) Interventions: patients in the experimental group were treated with CMI combined with WM or without WM, whereas the control group was treated with WM alone; (IV) Outcomes: RCTs contained one or more of the following outcome indicators: ① clinical effectiveness rate (CER = healing rate + markedly effective + effective rate), ②PaO2,③-PaCO2,④ diffusing capacity of the lungs for carbon monoxide (DLCO), ⑤FVC, ⑥TLC, ⑦ forced expiratory volume 1% (FEV1%), ⑧FEV1/FVC%, ⑨IIIC, and ⑩ transforming growth factor (TGF); and (V) The clinical efficacy criteria should meet the following requirements: good response or improvement: ① decrease in symptoms and increase in mobility, ② decrease in abnormal images on chest X-ray film or HRCT, ③ lung functional performance TLC, VC, DLCO, and PaO2 remained stable for a long time, with poor response or treatment failure; ① symptoms worsened, especially dyspnea and cough; ② increased abnormal images on chest X-ray or HRCT, especially in the cellular lung or pulmonary tissue. deterioration of lung function.

### 2.3 Data extraction and quality assessment

Two researchers independently screened the texts and extracted the data according to the established screening criteria, first reading the headings and abstracts, and then reading the relevant texts in full. The data extraction included the following: (I) basic information of the included literature (title, first author, date of publication, etc.); (II) basic information of the included patients (number of cases, age, gender, etc.); (III) interventions (drug name, dose, duration of disease, treatment course, etc.); (IV) outcome indicators (clinical efficacy, PaO2, DLCO, TGF-β1, etc.); (V) key elements of risk of bias evaluation (randomization method, blinding or not, allocation concealment, etc.). Finally, we used Excel to establish a data extraction table for extracting and summarizing the data from the included literature.

### 2.4 Risk of bias evaluation

The risk of bias of the included studies was assessed using the Cochrane Handbook's recommended risk of bias assessment tool, which covers seven areas: method of random sequence generation, whether allocation was hidden, whether patients and trial staff were blinded, whether outcome assessors were blinded, whether there were incomplete outcome data,

whether there was selective reporting, and other biases. The evaluation was then based on the actual situation of the included literature, with three categories of "low risk", "high risk", and "unclear".

### 2.5 Statistical analysis

We performed this study using RevMan (5.4) and Stata (16.0) and constructed a treatment strategy network. Dichotomous variables were expressed as the ORs, continuous variables as the mean difference (MD) or standardized mean difference (SMD), and 95% CIs were used for all interval estimates. When 95% CIs of MD or SMD did not include zero, the differences between the groups were considered statistically significant. Subsequently, we used the SUCRA to rank the efficacy of the treatment strategy for each outcome. When the SUCRA is closer to 100%, the effectiveness of the intervention under this outcome indicator is greater. (III) Stata software was used to process the outcome data and construct a network diagram of different interventions under the same outcome index.

## 3 Results

### 3.1 Study selection

A total of 5,743 studies were identified using the search strategy. A total of 2,301 duplicate reports, 436 reviews, 1,327 animal studies, and 895 clinical studies that did not conform to the CMIs interventions or were clearly diagnosed were excluded. Finally, 63 RCTs were included in the NMA. The research framework of the study is shown in Fig 1A and the selection process are illustrated in Fig 1B.

### 3.2 Characteristics of the included studies

This study involved a total of 14 CMIs combined with WM treatment strategies against IPF, including Shenfu injection (SF), Shenmai injection (SM), ligustrazine injection (LI), *Salvia miltiorrhiza* polyphenolate injection (SMP), DH, Xuebijing injection (XBJ), SX, SXT, Rhodiola injection (RI), Huangqi injection (HQ), safflower yellow sodium chloride injection (HHS), matrine injection (MI), SXN, and Guanxinning injection (GXN). The composition of the 14 CMIs is listed in Table 1. All RCTs were conducted in China, which spanned between 1998 and 2021. Among the participants in this study, approximately 73% were male, average age of the participants ranged from 3.87 to 71.23 years, and the duration of RCTs ranged from 2 to 24 weeks. Details of the study characteristics are presented in S1 Table [7–69]. The intervention in the control group was WM treatment, including glucocorticoid therapy, immunosuppressants, acetylcysteine, and other symptomatic Western medical treatments. The experimental group was treated with CMIs in addition to the control group. Stata 14.0 was used to create the web of relationships for each intervention, as shown in Fig 2.

### 3.3 Quality of the included studies

Among the included studies, 61 clearly reported the randomization method adopted, while two studies [62, 67] did not report the randomization method. The data of 63 studies were complete. We assessed the quality of the included studies using the Cochrane risk-of-bias tool. Each evaluation principle was divided into "high risk", "low risk", and "unclear" (Fig 3).

### 3.4. Outcomes of meta-analysis

No closed loops were included between interventions; therefore, there were no direct comparisons between interventions. Moreover, all pairwise comparisons between intervention drugs

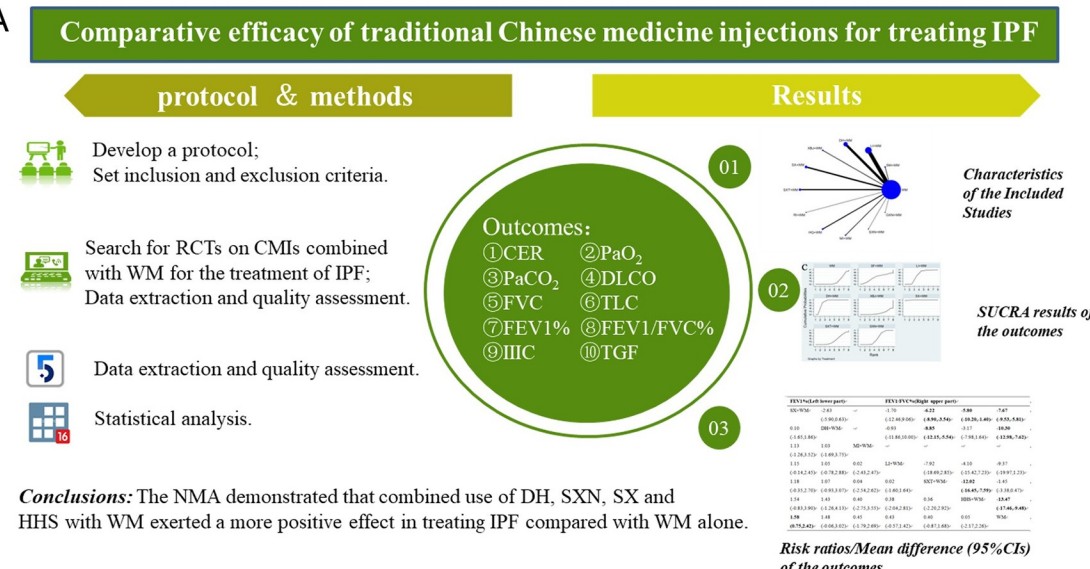

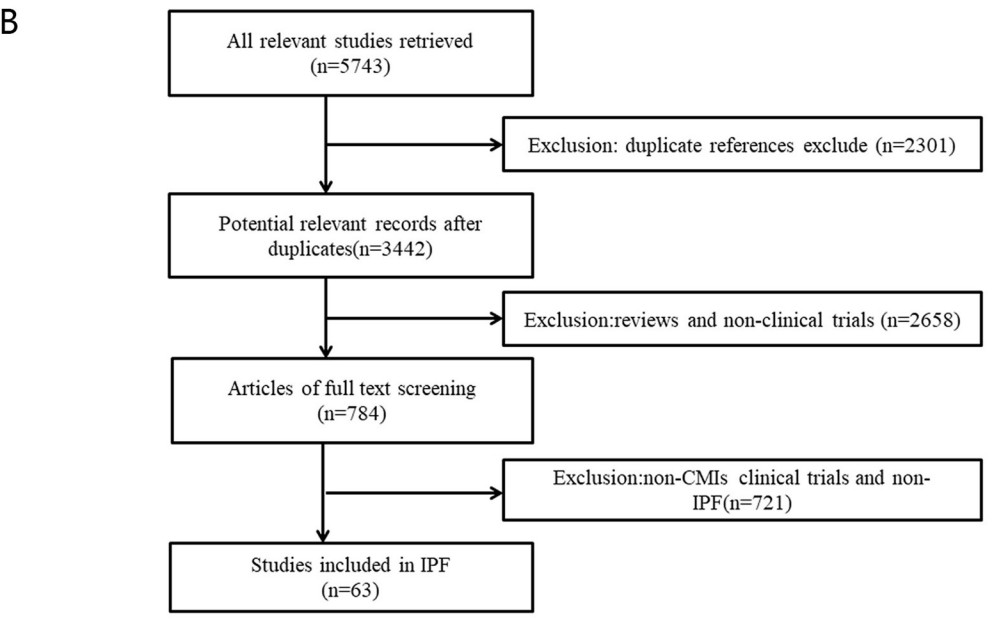

**Fig 1. A.** The research framework of the study. **B.** Flow chart of the search for eligible studies.

were indirect; therefore, no inconsistency tests were required. The results were analyzed directly using the consistency model.

**3.4.1. Effect on CER.** The primary outcome indicator in our study was CER. The results of the remaining 44 RCTs involving 11 types of CMIs were pooled to estimate the efficacy of the combination treatment in improving the partial pressure of oxygen and lung function, including PaO2, PaCO2, and DLCO (Fig 2A). The meta-analysis showed that the top five SUCRA rankings were SXN+WM (91.1), SXT+WM (85.6), SX+WM (75.5), DH+WM (61.4), and HQ+WM (53.2), as presented in Table 2 and Fig 4A. The higher the SUCRA value, the

**Table 1. Components of CMIs.**

| CMI | Chinese name | Species | Family |
|---|---|---|---|
| Shenfu Injection | Hongshen | *Talinum paniculatum* (Jacq.) Gaertn. | Araliaceae |
| | Fuzi | *Aconitum carmichaeli* Debx. | Ranunculaceae |
| Shenmai Injection | Hongshen | *Talinum paniculatum* (Jacq.) Gaertn. | Araliaceae |
| | Maidong | *Ophiopogon japonicus* (Linn. f.) Ker-Gawl. | Liliaceae |
| Ligustrazine Injection | Chuanxiong | *Ligusticum chuanxiong* Hort. | Umbelliferae |
| Salvia miltiorrhiza polyphenolate Injection | Danshen | *Salvia miltiorrhiza* Bge. | Lamiaceae |
| Danhong Injection | Danshen | *Salvia miltiorrhiza* Bge. | Lamiaceae |
| | Honghua | *Carthamus tinctorius* L. | Asteraceae |
| Xuebijing Injection | Honghua, | *Carthamus tinctorius* L. | Asteraceae |
| | Chishao | *Paeonia lactiflora* Pall. | Ranunculaceae |
| | Chuanxiong | *Ligusticum chuanxiong* Hort. | Umbelliferae |
| | Danshen | *Salvia miltiorrhiza* Bge. | Lamiaceae |
| | Danggui | *Angelica sinensis* (Oliv.) Diels | Umbelliferae |
| Shenxiong Injection | Danshen | *Salvia miltiorrhiza* Bge. | Lamiaceae |
| | Chuanxiong | *Ligusticum chuanxiong* Hort. | Umbelliferae |
| Shuxuetong Injection | Shuizhi | *Whitmania pigra* Whitman | Hirudinidae |
| | Dilong | *Pheretima aspergillum* (E. Perrier) | Megascolecidae |
| Rhodiola Injection | Hongjingtian | *Crassulaceae* J. St.-Hil. | Crassulaceae |
| Huangqi Injection | Huangqi | *Astragalus membranaceus* (Fisch.) Bunge | Leguminosae |
| safflower yellow sodium chloride injection | Honghua | *Carthamus tinctorius* L. | Asteraceae |
| Matrine Injection | Kushen | *Sophora flavescens* Ait. | Leguminosae |
| Shuxuening Injection | Yinxingye | *Ginkgo biloba* L. | Ginkgoales |
| Guanxinning Injection | Danshen | *Salvia miltiorrhiza* Bge. | Lamiaceae |
| | Chuanxiong | *Ligusticum chuanxiong* Hort. | Umbelliferae |

higher the likelihood that CMI will be a better intervention. Table 3 shows that SXN+WM (OR 8.91, 95% CI 3.81–20.83), SXT+WM (OR 7.36, 95% CI 3.30–16.00), SX+WM (OR 5.42, 95% CI 2.90–10.13), DH+WM (OR 4.06, 95% CI 2.62–6.29), HQ+WM (OR 3.47, 95% CI 1.55–7.77), LI+WM (OR 2.91, 95% CI 2.09–4.04), and SM+WM (OR 2.55, 95% CI 1.11–5.85) had a better clinical efficacy rate than that of WM alone, and the difference was statistically significant.

**3.4.2. Effect on PaO2, PaCO2, and DLCO.** A total of 39 RCTs reported PaO2 involving 11 types of CMIs (Fig 2B). The meta-analysis showed that the top five SUCRA rankings were HHS+WM (91.4), RI+WM (88.9), DH+WM (83.7), SX+WM (71.7), HQ+WM (55.6), and HHS+WM had the highest probability of being the best intervention for increasing PaO2 (Table 2 **and** Fig 4B). In addition, the OR value results showed that SXN+WM (OR -6.09, 95% CI -11.43,-0.75), SX+WM (OR -11.44, 95% CI -14.78,-8.11), DH+WM (OR -13.39, 95% CI -14.90,-11.89), HQ+WM (OR -8.70, 95% CI -15.91,-1.49), LI+WM (OR -5.33, 95% CI -7.67,-2.99), RI+WM (OR -15.44, 95% CI -23.22,-7.66), XBJ+WM (OR -6.80, 95% CI -12.37,-1.23), and HHS+WM (OR -8.85, 95% CI -17.05,-0.65) respectively (Table 3).

In terms of lung function in DLCO, 34 RCTs involving 11 types of CMIs were included (Fig 2D). The OR values showed that only LI+WM (OR -8.85, 95% CI -16.89,-0.81) had a statistically significant difference compared to that of WM alone, indicating that LI+WM had a good curative effect in improving DLCO (Table 4). The SUCRA values suggest that LI+WM (82.7) had the highest likelihood of being the best treatment for improving PaCO2 followed by SX+WM (Table 2 **and** Fig 4D).

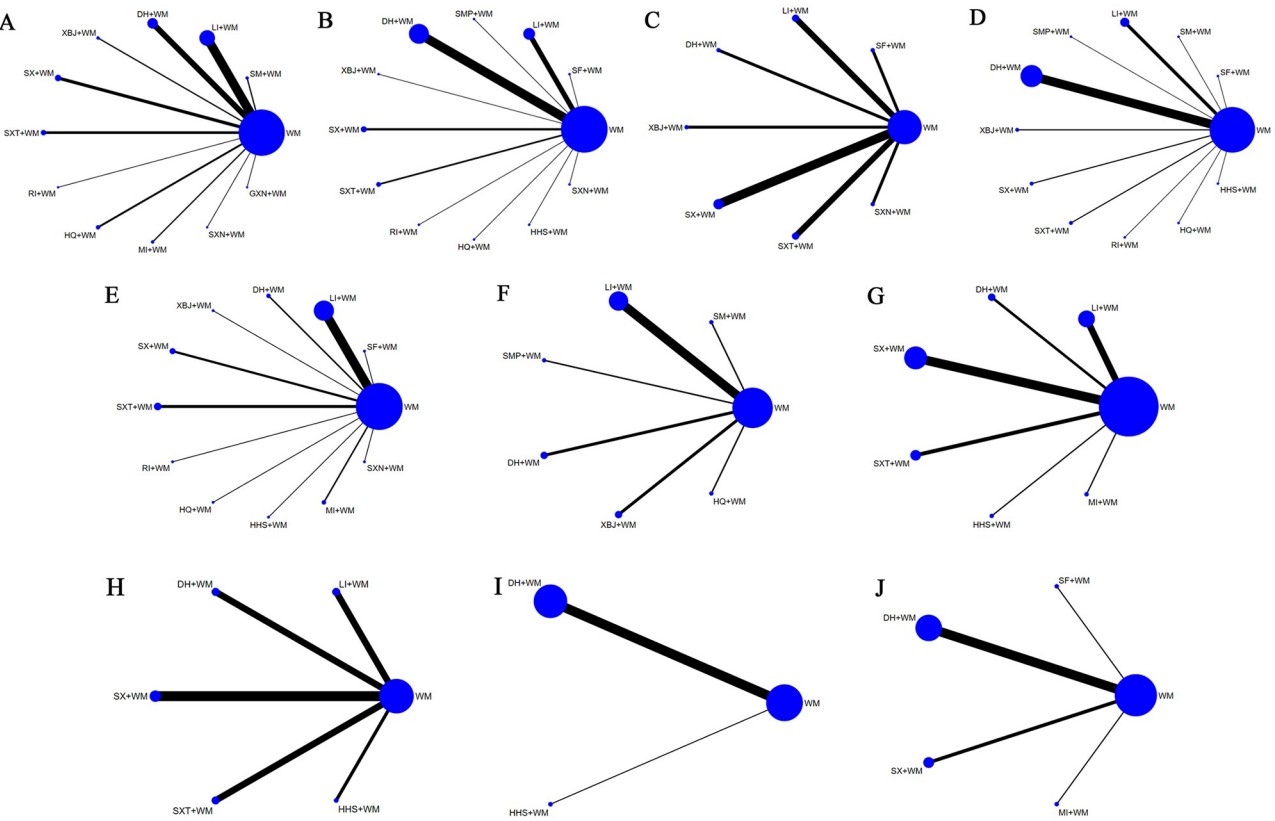

**Fig 2. Network relationship diagram of each outcome index of CMIs combined with WM in the treatment of IPF.** Note: **A**, CER; **B**, PaO2; **C**, PaCO2; **D**, DLCO; **E**, FVC; **F**, TLC; **G**, FEV1%; **H**, FEV1/FVC%; **I**, IIIC; and **J**, TGF.

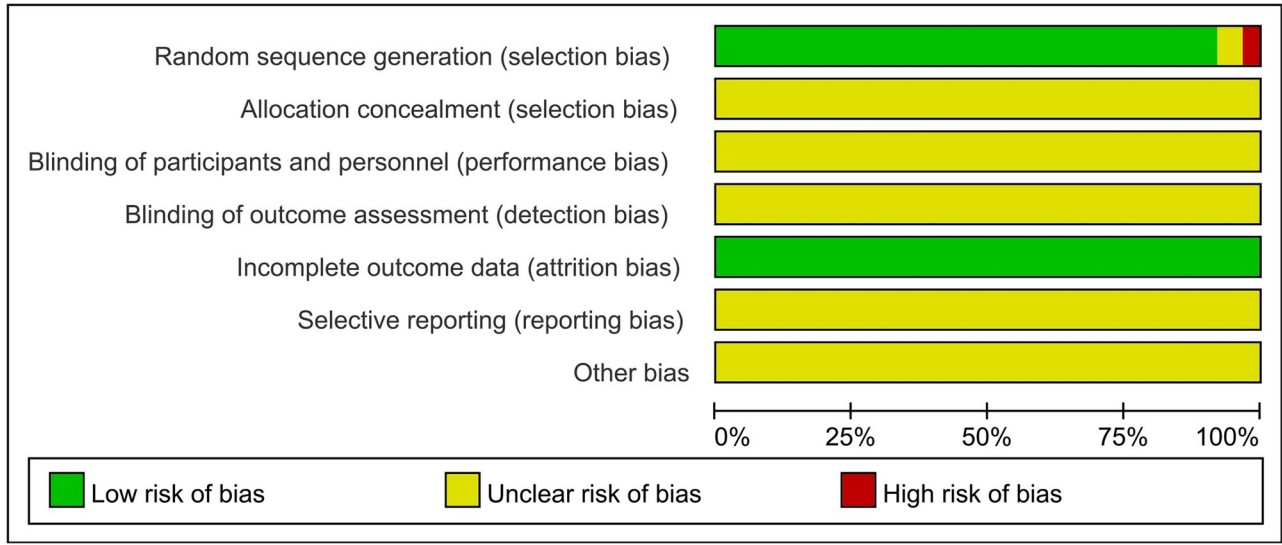

**Fig 3. Risk-of-bias graph.**

**Table 2. Surface under the cumulative ranking curve (SUCRA) results of the outcomes.**

| Intervention | CER | PaO2 | PaCO2 | DLCO | FVC | TLC | FEV1% | FEV1/FVC% | IIIC | TGF |
|---|---|---|---|---|---|---|---|---|---|---|
| SXN+WM | 91.1 | 40.4 | 45.4 | - | 99.9 | - | - | - | - | - |
| SXT+WM | 85.6 | 23.6 | 29.8 | 44.7 | 24.2 | - | 42.6 | 20.2 | - | - |
| SX+WM | 75.5 | 71.7 | 99.8 | 54.9 | 86.2 | - | 85.9 | 48.7 | - | 75.3 |
| DH+WM | 61.4 | 83.7 | 83.3 | 48.7 | 73.5 | 89.0 | 79.8 | 72.2 | 54.9 | 85.7 |
| HQ+WM | 53.2 | 55.6 | - | 47.6 | 41.7 | 45.3 | - | - | - | - |
| GXN+WM | 48.3 | | - | - | - | - | - | - | - | - |
| MI+WM | 47.9 | | - | - | 76.7 | - | 44.2 | - | - | 45.0 |
| LI+WM | 42.5 | 35.3 | 64.1 | 82.7 | 37.1 | 23.7 | 43.7 | 63.7 | - | - |
| SM+WM | 38.6 | | - | 45.3 | - | 23.6 | - | - | - | - |
| RI+WM | 34.5 | 88.9 | - | 46.3 | 34.1 | - | - | - | - | - |
| XBJ+WM | 15.9 | 44.7 | 11.1 | 45.7 | 36.6 | 67.0 | - | - | - | - |
| WM | 5.5 | 6.8 | 20.5 | 41.8 | 28.4 | 19.1 | 21.8 | 2.2 | 0.6 | 21.8 |
| SMP+WM | - | 47.0 | - | 46.9 | - | 82.4 | - | - | - | - |
| SF+WM | - | 10.9 | 45.9 | 45.6 | 28.7 | - | - | - | - | 22.2 |
| HHS+WM | - | 91.4 | - | 49.8 | 32.8 | - | 32.1 | 93.0 | 94.5 | - |

In terms of PaCO2, 11 RCTs involving 7 types of CMIs were included (Fig 2C). The SUCRA values suggested that SX+WM (99.8) was the highest followed by DH+WM (83.3) (Table 2 **and** Fig 4C). The results of NMA showed that SX+WM, DH+WM, and LI+WM could reduce PaCO2 compared to WM alone. The OR values and 95% CI values were SX +WM (OR -8.43, 95% CI -9.29,-7.57), DH+WM (OR -4.77, 95% CI -5.55,-3.99), LI+WM (OR -2.42,95% CI -4.36,-0.49), respectively (Table 5).

**3.4.3. Effect on FVC and TLC.** A total of 29 RCTs and 10 types of CMIs assessed FVC (Fig 2E). The SUCRA results were ranked as follows: SXN+WM (99.9) > SX+WM (86.2) > MI+WM (76.7) > DH+WM (73.5) (Table 2 **and** Fig 4E). Among them, their OR and 95% CI values were SXN+WM (OR -4.27, 95% CI -5.85,-2.69), SX+WM (OR -2.07, 95% CI -2.97,- 1.17), MI+WM (OR -1.56, 95% CI -2.71,-0.409), and DH+WM (OR -1.42, 95% CI -2.47,-0.36), all of which were statistically significant (Table 5).

Thirteen RCTs and 6 types of CMIs assessed TLC (Fig 2F). The SUCRA results were ranked as follows: DH+WM (89.0) > SMP+WM (82.4) > XBJ+WM (67.0) > HQ+WM (45.3) (Table 2 **and** Fig 4F). Table 4 shows that DH+WM (OR 0.93, 95% CI 0.51,1.36), SMP+WM (OR 0.86, 95% CI 0.17,1.56), and XBJ+WM (OR 0.58, 95% CI 0.06,1.11) had a more effective rate than that of WM alone, while the remaining three CMIs + WM compared with WM alone had no statistical significance.

**3.4.4. Effect on FEV1% and FEV1/FVC%.** Nineteen RCTs with six types of CMI intervention categories contributed to this NMA assessment of FEV1% (Fig 2G). The SUCRA values suggested that SX+WM (85.9) was the optimal choice for improving FEV1% levels in patients, followed by DH+WM (79.8) (Table 2 **and** Fig 4G). The OR value results showed that only SX+WM (OR 1.58, 95% CI 0.75,2.42) had a statistically significant difference compared with that of WM alone (Table 6).

Ten RCTs with five types of CMI intervention categories contributed to this NMA assessing the FEV1/FVC% (Fig 2H). The SUCRA values suggested that HHS + WM (93.0) had the highest likelihood of being the best treatment for improving FEV1/FVC%, followed by DH + WM (72.7) (Table 2 **and** Fig 4H). In IPF patients, SX+WM (OR -7.67, 95% CI -9.53,-5.81), DH +WM (OR -10.30, 95% CI -12.98,-7.62), and HHS+WM (OR -13.47, 95% CI -17.46,-9.48) significantly improved FEV1/FVC% (Table 6).

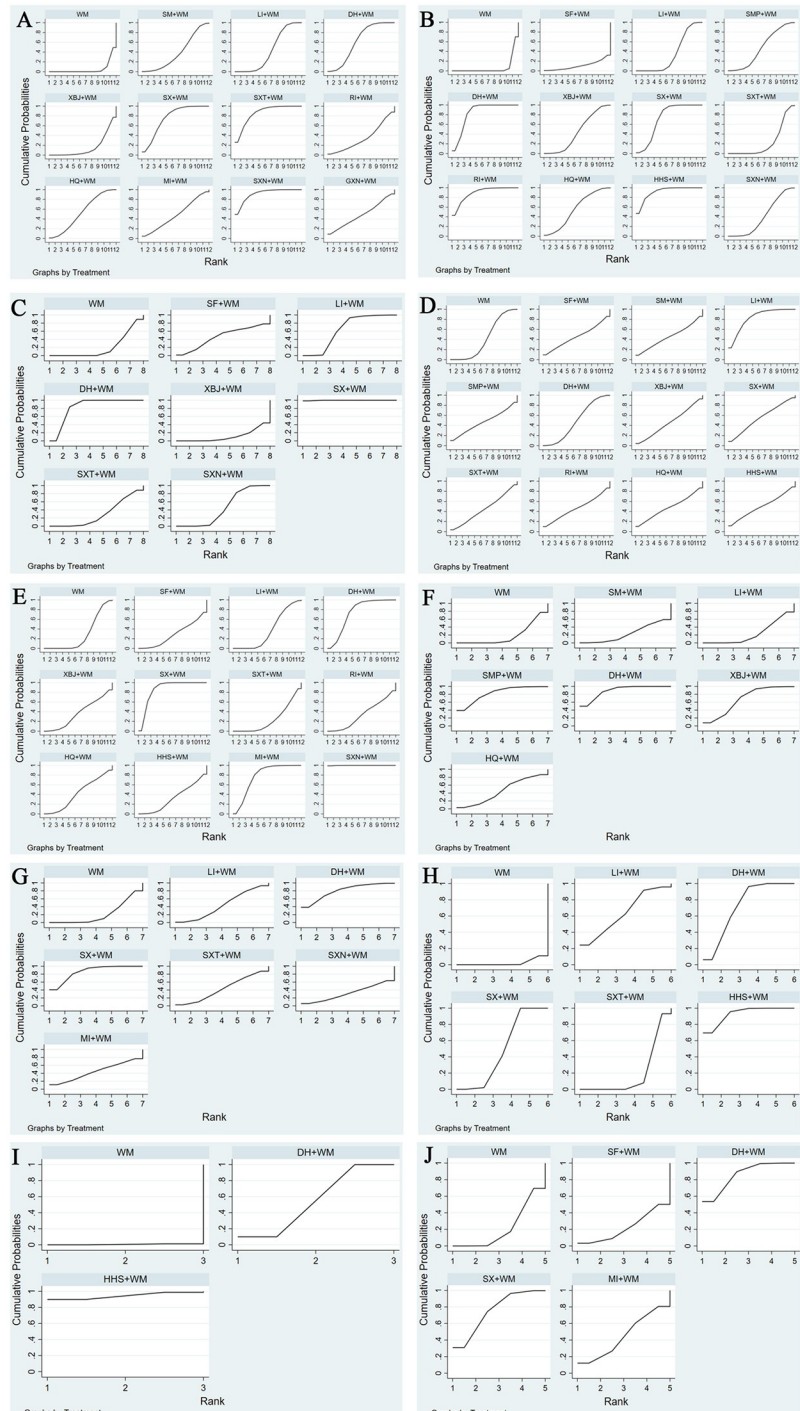

**Fig 4. Plot of the surface under the cumulative ranking curves for outcomes.** A, CER; B, PaO2; C, PaCO2; D, DLCO; E, FVC; F, TLC; G, FEV1%; H, FEV1/FVC%; I, IIIC; J, TGF.

**Table 3. Risk ratio/mean difference (95% CI) of the CER and PaO2.**

| CER (Left lower part) | | | | | | | | | | | PaO2 (Right upper part) | | | |
|---|---|---|---|---|---|---|---|---|---|---|---|---|---|---|
| SXN+WM | -2.66 (-9.07,3.74) | -5.35 (-11.65,0.95) | -7.30 (-12.86,-1.75) | -2.61 (-11.58,6.36) | - | - | -0.76 (-6.59,5.07) | - | -9.35 (-18.78,0.08) | -0.71 (-8.42,7.01) | -6.09 (-11.43,-0.75) | -9.56 (-17.61,-1.51) | -1.10 (-9.18,6.98) | -10.09 (-27.69,7.51) |
| 1.23 (0.38,3.91) | SXT+WM | -8.01 (-12.88,-3.15) | -9.97 (-13.82,-6.12) | -5.27 (-13.30,2.75) | - | - | -1.90 (-6.14,2.33) | - | -12.01 (-20.56,-3.47) | -3.37 (-9.97,3.22) | -3.43 (-6.96,0.11) | -12.22 (-19.21,-5.24) | -3.76 (-10.78,3.25) | -7.43 (-24.57,9.72) |
| 1.64 (0.57,4.72) | 1.34 (0.49,3.64) | SX+WM | -1.95 (-5.61,1.71) | -2.74 (-10.68,5.20) | - | - | -6.11 (-10.19,-2.03) | - | -4.00 (-12.46,4.46) | -4.64 (-11.13,1.85) | -11.44 (-14.78,-8.11) | -4.21 (-11.09,2.67) | -4.25 (-11.17,2.66) | -15.44 (-32.54,1.66) |
| 2.20 (0.84,5.71) | 1.79 (0.73,4.40) | 1.34 (0.63,2.84) | DH+WM | -4.69 (-12.06,2.67) | - | - | -8.06 (-10.85,-5.28) | - | -2.05 (-9.97,5.88) | -6.59 (-12.36,0.83) | -13.39 (-14.90,-11.89) | -2.26 (-8.46,3.95) | -6.20 (-12.45,0.04) | -17.39 (-34.24,-0.55) |
| 2.56 (0.80,8.27) | 2.09 (0.68,6.46) | 1.56 (0.56,4.32) | 1.17 (0.47,2.92) | HQ+WM | - | - | -3.37 (-10.95,4.21) | - | -6.74 (-17.34,3.86) | -1.90 (-11.01,7.21) | -8.70 (-15.91,-1.49) | -6.95 (-16.34,2.44) | -1.51 (-10.92,7.90) | -12.70 (-30.96,5.56) |
| 2.92 (0.42,20.37) | 2.38 (0.35,16.19) | 1.77 (0.28,11.36) | 1.33 (0.22,8.05) | 1.14 (0.17,7.79) | GXN+WM | - | - | - | - | - | - | - | - | - |
| 2.90 (0.58,14.40) | 2.36 (0.49,11.39) | 1.76 (0.39,7.88) | 1.32 (0.32,5.51) | 1.13 (0.23,5.49) | 0.99 (0.11,9.09) | MI+WM | - | - | - | - | - | - | - | - |
| 3.06 (1.23,7.61) | 2.50 (1.06,5.88) | 1.86 (0.92,3.79) | 1.39 (0.80,2.42) | 1.19 (0.50,2.85) | 1.05 (0.18,6.21) | 1.06 (0.26,4.28) | LI+WM | - | -10.11 (-18.23,-1.99) | -1.47 (-7.51,4.57) | -5.33 (-7.67,-2.99) | -10.32 (-16.78,-3.86) | -1.86 (-8.35,4.63) | -9.33 (-26.27,7.60) |
| 3.49 (1.06,11.45) | 2.84 (0.91,8.93) | 2.12 (0.76,5.97) | 1.59 (0.62,4.06) | 1.36 (0.43,4.33) | 1.20 (0.17,8.28) | 1.20 (0.24,5.93) | 1.14 (0.47,2.79) | SM+WM | - | - | - | - | - | - |
| 4.14 (0.78,21.86) | 3.37 (0.66,17.29) | 2.52 (0.53,12.00) | 1.88 (0.42,8.42) | 1.61 (0.31,8.34) | 1.42 (0.15,13.57) | 1.43 (0.20,10.28) | 1.35 (0.31,5.87) | 1.18 (0.23,6.20) | RI+WM | -8.64 (-18.20,0.92) | -15.44 (-23.22,-7.66) | -0.21 (-10.05,9.63) | -8.25 (-18.11,1.61) | -19.44 (-37.93,-0.95) |
| 6.73 (1.87,24.25) | 5.48 (1.58,19.07) | 4.09 (1.29,12.98) | 3.06 (1.06,8.84) | 2.62 (0.75,9.19) | 2.30 (0.31,16.93) | 2.32 (0.44,12.26) | 2.20 (0.80,6.06) | 1.93 (0.54,6.87) | 1.63 (0.29,9.12) | XBJ+WM | -6.80 (-12.37,-1.23) | -8.85 (-17.05,0.65) | -0.39 (-8.62,7.84) | -10.80 (-28.47,6.87) |
| 8.91 (3.81,20.83) | 7.26 (3.30,16.00) | 5.42 (2.90,10.13) | 4.06 (2.62,6.29) | 3.47 (1.55,7.77) | 3.05 (0.53,17.52) | 3.07 (0.79,11.97) | 2.91 (2.09,4.04) | 2.55 (1.11,5.85) | 2.15 (0.51,9.02) | 1.32 (0.51,3.46) | WM | -15.65 (-21.67,-9.63) | -7.19 (-13.25,1.13) | 4.00 (-12.77,20.77) |
| - | - | - | - | - | - | - | - | - | - | - | - | HHS+WM | -8.46 (-17.00,0.08) | -19.65 (-37.47,-1.83) |
| - | - | - | - | - | - | - | - | - | - | - | - | - | SMP+WM | -11.19 (-29.02,6.64) |
| - | - | - | - | - | - | - | - | - | - | - | - | - | - | SF+WM |

**Table 4. Risk ratios/Mean difference (95%CIs) of the TLC and DLCO.**

TLC (Left lower part) — lower-left triangle; DLCO (Right upper part) — upper-right triangle. Diagonal cells hold treatment names; values below a diagonal name are TLC comparisons, values above are DLCO comparisons.

| | LI+WM | SX+WM | HHS+WM | DH+WM | HQ+WM | SMP+WM | RI+WM | XBJ+WM | SF+WM | SM+WM | SXT+WM | WM |
|---|---|---|---|---|---|---|---|---|---|---|---|---|
| **LI+WM** | LI+WM | -6.36 (-22.22,9.50) | -7.46 (-28.39,13.47) | -7.84 (-17.21,1.54) | -8.60 (-29.53,12.34) | -8.29 (-29.22,12.64) | -8.33 (-29.26,12.61) | -8.48 (-24.34,7.37) | -8.90 (-29.84,12.05) | -8.87 (-29.80,12.06) | -8.83 (-24.69,7.02) | **-8.85 (-16.89,-0.81)** |
| **SX+WM** | - | SX+WM | -1.10 (-24.77,22.57) | -1.48 (-15.98,13.02) | -2.24 (-25.91,21.44) | -1.93 (-25.60,21.74) | -1.97 (-25.64,21.71) | -2.12 (-21.45,17.21) | -2.54 (-26.22,21.15) | -2.51 (-26.18,21.16) | -2.47 (-21.80,16.86) | -2.49 (-16.16,11.18) |
| **HHS+WM** | - | - | HHS+WM | -0.37 (-20.30,19.55) | -1.14 (-28.47,26.20) | -0.83 (-28.16,26.50) | -0.86 (-28.20,26.47) | -1.02 (-24.69,22.65) | -1.44 (-28.78,25.90) | -1.41 (-28.74,25.92) | -1.37 (-25.04,22.30) | -1.39 (-20.72,17.94) |
| **DH+WM** | **0.90 (0.39,1.42)** | **0.93 (0.12,1.74)** | - | DH+WM | -0.76 (-20.68,19.16) | -0.45 (-20.37,19.47) | -0.49 (-20.41,19.43) | -0.65 (-15.14,13.85) | -1.06 (-20.99,18.87) | -1.03 (-20.95,18.89) | -1.00 (-15.49,13.50) | -1.02 (-5.85,3.82) |
| **HQ+WM** | 0.27 (-0.53,1.07) | 0.64 (-0.23,1.50) | - | HQ+WM | HQ+WM | 0.31 (-27.02,27.64) | 0.27 (-27.06,27.61) | 0.12 (-23.56,23.79) | -0.30 (-27.64,27.04) | -0.27 (-27.60,27.06) | -0.24 (-23.91,23.44) | -0.25 (-19.58,19.07) |
| **SMP+WM** | **0.83 (0.08,1.59)** | 0.57 (-0.46,1.59) | SMP+WM | - | - | SMP+WM | -0.04 (-27.37,27.29) | -0.19 (-23.86,23.47) | -0.61 (-27.95,26.73) | -0.58 (-27.91,26.75) | -0.55 (-24.21,23.12) | -0.56 (-19.89,18.76) |
| **RI+WM** | - | - | - | - | - | RI+WM | RI+WM | -0.16 (-23.83,23.52) | -0.57 (-27.91,26.77) | -0.54 (-27.88,26.79) | -0.51 (-24.18,23.16) | -0.52 (-19.85,18.80) |
| **XBJ+WM** | 0.55 (-0.05,1.15) | 0.35 (-0.33,1.03) | - | - | - | - | XBJ+WM | XBJ+WM | -0.41 (-24.10,23.27) | -0.39 (-24.06,23.28) | -0.35 (-19.68,18.98) | -0.37 (-14.03,13.30) |
| **SF+WM** | - | - | - | - | - | - | - | SF+WM | SF+WM | 0.03 (-27.31,27.37) | 0.06 (-23.62,23.74) | 0.05 (-19.29,19.39) |
| **SM+WM** | 0.03 (-0.72,0.77) | **0.93 (0.12,1.74)** | 0.30 (-0.72,1.31) | - | - | - | - | - | - | SM+WM | 0.04 (-23.63,23.70) | 0.02 (-19.31,19.34) |
| **SXT+WM** | - | - | - | - | - | - | - | - | - | - | SXT+WM | -0.02 (-13.68,13.65) |
| **WM** | 0.03 (-0.25,0.32) | **0.93 (0.51,1.36)** | 0.30 (-0.45,1.05) | **0.86 (0.17,1.56)** | 0.28 (-0.59,1.16) | **0.58 (0.06,1.11)** | 0.31 (-0.46,1.59) | 0.28 (-0.63,1.20) | - | 0.00 (-0.69,0.69) | - | WM |

**Table 5. Risk ratios/Mean difference (95%CIs) of the PaCO2 and FVC.**

| PaCO2 (Left lower part) | | | | | | FVC (Right upper part) | | | | | |
|---|---|---|---|---|---|---|---|---|---|---|---|
| SXN+WM | -2.20 (-4.02,-0.39) | -2.72 (-4.68,-0.76) | -2.86 (-4.76,-0.96) | -3.96 (-6.17,-1.75) | -4.13 (-5.77,-2.49) | -4.10 (-6.33,-1.86) | -4.19 (-6.41,-1.96) | -4.21 (-6.42,-2.00) | -4.37 (-6.68,-2.06) | -4.27 (-5.85,-2.69) | -4.40 (-6.16,-2.64) |
| -7.53 (-8.45,-6.62) | SX+WM | -0.51 (-1.98,0.95) | -0.66 (-2.04,0.73) | -1.76 (-3.55,0.04) | -1.93 (-2.93,-0.92) | -1.89 (-3.71,-0.08) | -1.98 (-3.79,-0.18) | -2.00 (-3.79,-0.22) | -2.17 (-4.08,-0.26) | -2.07 (-2.97,-1.17) | -2.19 (-3.38,-1.01) |
| - | - | MI+WM | -0.14 (-1.71,1.43) | -1.24 (-3.17,0.69) | -1.41 (-2.65,-0.17) | -1.38 (-3.34,0.58) | -1.47 (-3.42,0.48) | -1.49 (-3.42,0.44) | -1.65 (-3.70,0.39) | -1.56 (-2.71,-0.40) | -1.68 (-3.07,-0.29) |
| -3.87 (-4.71,-3.03) | -3.66 (-4.82,-2.50) | - | DH+WM | -1.10 (-2.98,0.78) | -1.27 (-2.42,-0.12) | -1.24 (-3.14,0.66) | -1.33 (-3.22,0.56) | -1.35 (-3.22,0.52) | -1.51 (-3.50,0.47) | -1.42 (-2.47,-0.36) | -1.54 (-2.85,-0.23) |
| - | - | - | - | HQ+WM | -0.17 (-1.78,1.44) | -0.14 (-2.35,2.08) | -0.23 (-2.43,1.97) | -0.25 (-2.44,1.94) | -0.41 (-2.70,1.87) | -0.32 (-1.87,1.23) | -0.44 (-2.17,1.29) |
| -1.52 (-3.49,0.44) | -6.01 (-8.13,-3.89) | - | -2.35 (-4.43,-0.26) | - | LI+WM | 0.03 (-1.61,1.67) | -0.06 (-1.69,1.57) | -0.08 (-1.68,1.53) | -0.24 (-1.98,1.50) | -0.14 (-0.59,0.30) | -0.27 (-1.16,0.62) |
| -1.70 (-3.91,0.51) | -9.23 (-11.58,-6.89) | - | -5.57 (-7.89,-3.25) | - | -3.22 (-6.14,-0.30) | XBJ+WM | -0.09 (-2.32,2.13) | -0.11 (-2.32,2.10) | -0.28 (-2.59,2.03) | -0.18 (-1.76,1.40) | -0.30 (-2.06,1.46) |
| - | - | - | - | - | - | - | RI+WM | -0.02 (-2.22,2.18) | -0.19 (-2.48,2.11) | -0.09 (-1.65,1.48) | -0.21 (-1.96,1.54) |
| - | - | - | - | - | - | - | - | HHS+WM | -0.17 (-2.45,2.12) | -0.07 (-1.61,1.48) | -0.19 (-1.92,1.54) |
| -0.60 (-6.77,5.57) | -6.93 (-13.16,-0.71) | - | -3.27 (-9.49,2.95) | - | -0.92 (-7.39,5.54) | -2.30 (-8.84,4.24) | - | - | SF+WM | 0.10 (-1.59,1.78) | -0.02 (-1.88,1.83) |
| -0.90 (-1.21,-0.59) | -8.43 (-9.29,-7.57) | - | -4.77 (-5.55,-3.99) | - | -2.42 (-4.36,-0.49) | -0.80 (-2.98,1.38) | - | - | -1.50 (-7.67,4.67) | WM | -0.12 (-0.90,0.65) |
| -0.55 (-2.24,1.13) | -8.09 (-9.95,-6.22) | - | -4.42 (-6.25,-2.60) | - | -2.08 (-4.62,0.47) | -1.15 (-3.89,1.59) | - | - | -1.15 (-7.54,5.23) | -0.35 (-2.00,1.31) | SXT+WM |

**Table 6. Risk ratio/mean difference (95% CI) of the FEV1% and FEV1/FVC%.**

| FEV1% (Left lower part) | | | FEV1/FVC% (Right upper part) | | | |
|---|---|---|---|---|---|---|
| SX+WM | -2.63 (-5.90,0.63) | - | -1.70 (-12.46,9.06) | -6.22 (-8.90,-3.54) | -5.80 (-10.20,-1.40) | -7.67 (-9.53,-5.81) |
| 0.10 (-1.65,1.86) | DH+WM | - | -0.93 (-11.86,10.00) | -8.85 (-12.15,-5.54) | -3.17 (-7.98,1.64) | -10.30 (-12.98,-7.62) |
| 1.13 (-1.26,3.52) | 1.03 (-1.69,3.75) | MI+WM | - | - | - | - |
| 1.15 (-0.14,2.45) | 1.05 (-0.78,2.88) | 0.02 (-2.43,2.47) | LI+WM | -7.92 (-18.69,2.85) | -4.10 (-15.42,7.23) | -9.37 (-19.97,1.23) |
| 1.18 (-0.35,2.70) | 1.07 (-0.93,3.07) | 0.04 (-2.54,2.62) | 0.02 (-1.60,1.64) | SXT+WM | -12.02 (-16.45,-7.59) | -1.45 (-3.38,0.47) |
| 1.54 (-0.83,3.90) | 1.43 (-1.26,4.13) | 0.40 (-2.75,3.55) | 0.38 (-2.04,2.81) | 0.36 (-2.20,2.92) | HHS+WM | -13.47 (-17.46,-9.48) |
| 1.58 (0.75,2.42) | 1.48 (-0.06,3.02) | 0.45 (-1.79,2.69) | 0.43 (-0.57,1.42) | 0.40 (-0.87,1.68) | 0.05 (-2.17,2.26) | WM |

Efficacy of traditional Chinese medicine injections for treating idiopathic pulmonary fibrosis

**Table 7. Risk ratio/mean difference (95% CI) of the TGF and IIIC.**

| TGF (Left lower part) | | | | IIIC (Right upper part) | |
|---|---|---|---|---|---|
| DH+WM | - | - | - | **13.08 (5.11,21.05)** | 17.62 (-9.24,44.48) |
| -0.65 (-4.18,2.88) | SX+WM | - | - | - | - |
| -2.74 (-8.24,2.76) | -2.09 (-8.09,3.91) | MI+WM | - | - | - |
| -4.64 (-10.17,0.89) | -3.99 (-10.01,2.03) | -1.90 (-9.25,5.45) | SF+WM | - | - |
| **-4.22 (-6.06,-2.37)** | **-3.57 (-6.58,-0.56)** | -1.48 (-6.66,3.71) | 0.42 (-4.79,5.63) | WM | **30.70 (5.05,56.35)** |
| - | - | - | - | - | HHS+WM |

**3.4.5. Effect on IIIC and TGF.** Ten RCTs involving two types of CMIs reported IIIC and were included in the NMA (Fig 2I). According to the SUCRA values, HHS+WM (94.5) had the highest probability of being the best treatment for reducing IIIC levels (Table 2 **and** Fig 4I). The interventions that were found to significantly reduce IIIC levels when compared with WM alone were HHS+WM (OR 30.70, 95% CI 5.05,56.35) and DH+WM (OR 13.08, 95% CI 5.11,21.05) (Table 7).

Thirteen RCTs involving 4 types of CMIs reported TGF and were included in the NMA (Fig 2J). According to the SUCRA values, DH+WM (85.7) had the highest probability of being the best treatment for reducing TGF levels (Table 2 **and** Fig 4J). SX+WM (OR -3.57, 95% CI -6.58,-0.56) and DH+WM (OR -4.22, 95% CI -6.06,-2.37) were significantly more effective in reducing TGF levels than WM alone (Table 7).

## 3.5. Publication bias

Publication bias was assessed for the above-mentioned 10 outcomes using funnel plots. Points with different colors represent different comparisons between the interventions. Fig 5 shows that the funnel plots were not visually symmetrical, indicating the existence of bias.

## 4 Discussion

IPF is a progressive pulmonary interstitial inflammatory disease and a sequela of severe patients with novel coronavirus pneumonia in 2019 [70]. The prevalence of IPF is increasing significantly, and the treatment options for IPF are very limited and ineffective due to the pathogenesis of IPF is not clear. Researchers have proposed some effective methods to treat IPF. WM treatment of IPF mainly involved pharmacological treatment and non-pharmacological treatment (e.g. lung transplantation). However, the long-term use of high doses of glucocorticoids can cause more harm than good, with many irreversible side effects and adverse reactions that can cause irreversible damage to the body [71]. Therefore, there is considerable debate in the medical community as to whether hormone therapy should be used long-term in patients with IPF.

TCM has a long history in the treatment of IPF, and treatment is mainly based on syndrome differentiation [72]. Combinations of CMIs and WM have been widely used to treat IPF, increase CER, and improve patients' quality of life. However, there is a lack of comprehensive and systematic evidence to support the beneficial effects of CMI in combination with WM in IPF.

In this study, NMA was used to systematically evaluate the efficacy of CMIs combined with WM in the treatment of IPF. Meanwhile, WM treatment of IPF mainly involves

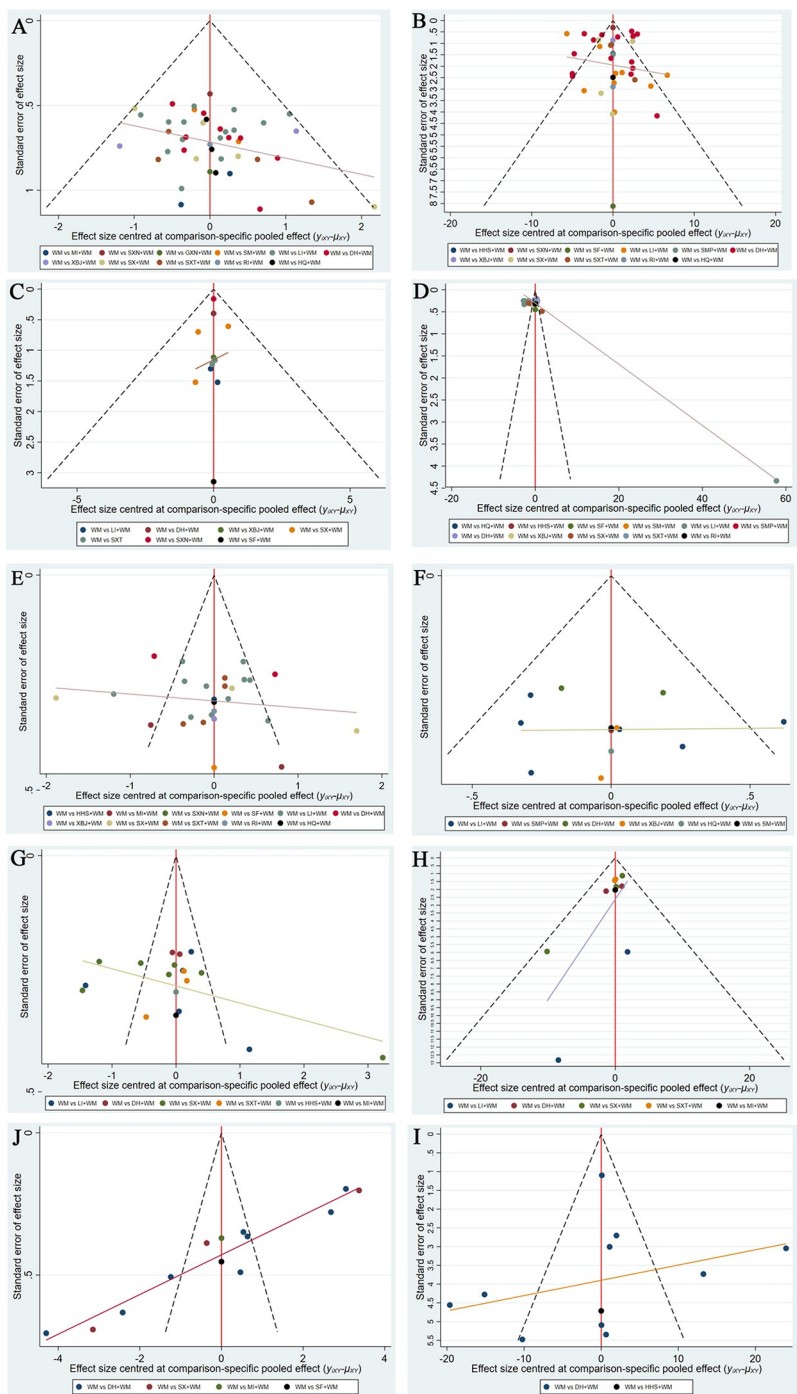

**Fig 5. Funnel plots.** A, CER; B, PaO2; C, PaCO2; D, DLCO; E, FVC; F, TLC; G, FEV1%; H, FEV1/FVC%; I, IIIC; J, TGF.

pharmacological treatment (e.g. glucocorticoids, anti-fibrotic drugs) and non-pharmacological treatment (e.g. oxygen and ventilators). Different outcome indicators such as CER, PaO2, PacO2, pulmonary function test (DLCO, FVC, TLC, FEV1%, and FEV1/FVC%), and molecular composition (IIIC and TGF)-associated fibrosis were analyzed to identify drugs with more

efficacy advantages and provide a reference basis for the selection of CMIs for patients with IPF. A total of 63 RCTs involving 14 types of CMIs were eventually included in this NMA for data analysis. As shown in Table 2, the SUCRA of DH+WM in CER, PaO2, PaCO2, TLC, and TGF were 61.4%, 83.7%, 83.3%, 89.0%, and 85.7%, respectively, ranking in the top 3. In addition, the SUCRA of SX+WM in CER, PaO2, PaCO2, FVC, FEV1%, and TGF were 75.5%, 71.7%, 99.8%, 86.2%, 85.9%, and 75.3%, ranking in the top 3. In terms of CER, SXN+WM, SXT+WM, SX+WM, DH+WM, and HQ+WM were the top five treatment strategies with the highest probability of being the best intervention. Furthermore, SXN combined with WM had the best performance in the CER (91.1%) and FVC (99.9%); HHS combined with WM had the best performance in the PaO2 (91.4%), FEV1/FVC% (93.0%), and IIIC (94.5%), all of which are important options for the treatment of IPF.

IPF belongs to the category of "lung atrophy", "lung arthralgia", or "asthma syndrome" in TCM disease names, whose pathology is characterized by a combination of "deficiency and reality". The "deficiency" focuses on the deficiencies of the lungs, spleen and kidneys, mostly in the qi and yin; the "reality" is based on pathogenic stagnation, dryness and heat, qi knots, phlegm, blood stasis and toxicity, with phlegm and stasis blocking being particularly critical to the disease. Treatment is based on "resolving phlegm and clearing the channels, activating blood stasis, and detoxifying and dispersing nodules".

Most CMIs have the ability to invigorate blood circulation to resolve blood stasis and calm asthma. Patients with IPF have severe pulmonary fibrosis, mainly related to alveolar damage, repair, and fibrotic processes, and therefore, show a significant increase in serum fibrosis markers. Previous studies have found that DH can effectively improve microcirculatory ischemia and hypoxia in IPF patients by increasing tissue-type fibrinogen activator activity and inhibiting fibrinogen activator inhibitor-I activation, platelet aggregation, and hypoxia inducible factor1 (HIF-1) activity [64]. HHS can promote fibrin degradation and fibrotic damage within lung tissue, and delay the decline of lung function by reducing the levels of hyaluronic acid (HA), laminin (LN), and type III procollagen (PCIII) [16]. The main ingredient of SXN is *Ginkgo biloba* extract, which functions in "astringing lung qi, calming asthma and cough, and stopping turbidity". The flavonoids in *Ginkgo biloba* extract can significantly increase blood oxygen concentration, which has an inhibitory effect on serum interleukin- 6 (IL6), IL8, and tumor necrosis factor-α (TNF-α) levels in patients with IPF [38]. The main constituents of SX are Danshen and Chuanxiong, both of which improve microcirculation and inhibit excessive inflammation [27]. LI can excite the respiratory and vasomotor centers of the medulla oblongata, relieve bronchospasm, inhibit platelet agglutination, reduce blood viscosity, and have a good inhibitory effect on elastase; therefore, it can effectively prevent the destruction of elastic fibers in the lungs, improve blood rheological indices and microcirculation, increase myocardial contractility, and improve ischemia and hypoxia in all vital organs [9].

This study has certain drawbacks that are worth mentioning; there is instability and bias in the results of this study in addition to limitations in the quality of RCTs. Moreover, the generalizability of the results is reduced since all the included RCTs were conducted in China. However, this study provides clinicians with detailed comparisons of common therapeutic strategies and a reference for clinical applications. The effectiveness and safety of CMIs in the treatment of IPF need to be further validated in future clinical studies. These findings need to be further validated by more large-scale studies with high quality and normative reporting.

## 5 Conclusions

IPF is a slowly progressive lung disease with a high mortality rate and an unknown pathogenesis. There is no effective cure for IPF, and CMIs combined with WM is expected to be one of

the future strategies for IPF treatment. This NMA confirms that the 14 treatment strategies used in this study improved the symptoms and clinical indicators of patients with IPF in different ways. The combined use of DH, SXN, SX, and HHS with WM exerted a more positive effect in treating IPF than that done by WM alone. However, it is worth noting that since different types of CMIs have different effects and functions in patients, patients' conditions should also be considered while selecting the CMIs to play a better role in the treatment of IPF.

## Supporting information

**S1 File. PRISMA NMA checklist.**
(DOCX)

**S2 File. The protocol of this review.**
(PDF)

**S3 File. Search strategy.**
(DOCX)

**S4 File. Summary of abbreviations in text.**
(DOCX)

**S1 Table. Characteristics of the included studies.**
(DOCX)

## Author Contributions

**Conceptualization:** Hong-sheng Cui.

**Data curation:** Shuai-yang Huang, Ming-sheng Lyu, Gui-rui Huang.

**Formal analysis:** Shuai-yang Huang, Ming-sheng Lyu.

**Funding acquisition:** Hong-sheng Cui.

**Investigation:** Ming-sheng Lyu, Gui-rui Huang, Dan Hou.

**Methodology:** Shuai-yang Huang, Gui-rui Huang, Dan Hou, Ming-xia Yu.

**Software:** Gui-rui Huang, Dan Hou, Ming-xia Yu.

**Writing – original draft:** Shuai-yang Huang.

**Writing – review & editing:** Shuai-yang Huang, Hong-sheng Cui, Ming-sheng Lyu.

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
