## [Decision Letter · Decision Letter 0]

27 Jun 2022

PONE-D-22-09987Efficacy of traditional Chinese medicine injections for treating idiopathic pulmonary fibrosis: a systematic review and network meta-analysisPLOS ONE

Dear Dr. Cui,

Thank you for submitting your manuscript to PLOS ONE. After careful consideration, we feel that it has merit but does not fully meet PLOS ONE’s publication criteria as it currently stands. Therefore, we invite you to submit a revised version of the manuscript that addresses the points raised during the review process.

We look forward to receiving your revised manuscript.

Kind regards,

Jen-Tsung Chen, Ph.D.

Academic Editor

PLOS ONE

Journal Requirements:

6. Please upload a new copy of Figure 5 as the detail is not clear. Please follow the link for more information: https://blogs.plos.org/plos/2019/06/looking-good-tips-for-creating-your-plos-figures-graphics/

Reviewers' comments:

Reviewer's Responses to Questions

**Comments to the Author**

1. Is the manuscript technically sound, and do the data support the conclusions?

Reviewer #1: Yes

Reviewer #2: Partly

2. Has the statistical analysis been performed appropriately and rigorously? 

Reviewer #1: I Don't Know

Reviewer #2: Yes

3. Have the authors made all data underlying the findings in their manuscript fully available?

Reviewer #1: Yes

Reviewer #2: Yes

4. Is the manuscript presented in an intelligible fashion and written in standard English?

Reviewer #1: Yes

Reviewer #2: Yes

5. Review Comments to the Author

Reviewer #1: The article "Efficacy of traditional Chinese medicine injections for treating idiopathic pulmonary fibrosis: a systematic review and network meta-analysis" is well organized . They have extensively studied available randomized trials evaluating the efficacy of co-administration of TCM and WM over WM. Among the available TCMs, DH+WM have been reported with better efficacy. In the discussion part, I would suggest the authors to explain the types of western medicines studied in randomized controlled trials selected for this study.

Reviewer #2: The manuscript entitled “Efficacy of traditional Chinese medicine injections for treating idiopathic pulmonary fibrosis: a systematic review and network meta-analysis”. Authors aimed to systematically evaluate the efficacy of Chinese medicine injections (CMI) in combination with Western medicine. The study included 14 CMIs applied in the clinical treatment of Idiopathic pulmonary fibrosis.

I recommend this manuscript for the reconsideration for publication in PlosOne after incorporating Major changes given in below.

COMMENTS:

Authors must concentrate on the formatting, and use of symbols, etc., throughout the manuscript.

Abstract should be revised and it should be more focus on your aim and results.

Introduction and last paragraph pf the introduction section should be elaborated in detail for better understanding of your studied issue.

More abbreviation makes difficult to understand the concept. Please revise it.

Framework figure is required. It will be useful to the readers for better understanding of the studied issue.

In conclusion section authors should provide some future prospectus related to present study.

Authors should clearly state the figure caption as well as description for ease of understanding of the readers.

6. PLOS authors have the option to publish the peer review history of their article (what does this mean?). If published, this will include your full peer review and any attached files.

Reviewer #1: No

Reviewer #2: No

---

## [Author Response · Author response to Decision Letter 0]

5 Jul 2022

Dea Reviewers,

Thank you for your letter and the reviewers recommend reconsideration of our manuscript entitled “Efficacy of traditional Chinese medicine injections for treating idiopathic pulmonary fibrosis: a systematic review and network meta-analysis” (Manuscript Number: PONE-D-22-09987). These comments are all valuable and very helpful for revising and improving this paper, as well as the important guiding significance to our research. We revised it completely according to the suggestions of reviewers, and supplemented pictures, texts and references. We hope the revised manuscript could meet the reviewer’s comments and suitable for publication in PLOS ONE. Revised portions are marked in Red Font in the revised manuscript

Reviewer: 1

#1. In the discussion part, I would suggest the authors to explain the types of western medicines studied in randomized controlled trials selected for this study.

Response: Thank you for your suggestion. We performed a literature research, and added the types of western medicines studied in randomized controlled trials selected for this study. Please see below. Thanks again for your advice.

IPF is a progressive pulmonary interstitial inflammatory disease and a sequela of severe patients with novel coronavirus pneumonia in 2019 [69]. The prevalence of IPF is increasing significantly, and the treatment options for IPF are very limited and ineffective due to the pathogenesis of IPF is not clear. Researchers have proposed some effective methods to treat IPF. WM treatment of IPF mainly involved pharmacological treatment and non-pharmacological treatment (e.g. lung transplantation). However, the long-term use of high doses of glucocorticoids can cause more harm than good, with many irreversible side effects and adverse reactions that can cause irreversible damage to the body[70]. Nedanib and pirfenidone are the only two known drugs recommended by the US Food and Drug Administration for the treatment of IPF. Nonetheless, these drugs are not only expensive, but their clinical efficacy is uncertain and they often cause adverse side effects during clinical use, most commonly gastrointestinal disorders, including diarrhoea, nausea and vomiting[4]. Therefore, there is considerable debate in the medical community as to whether hormone therapy should be used long-term in patients with IPF. TCM has a long history in the treatment of IPF, and treatment is mainly based on syndrome differentiation [71]. Combinations of CMIs and WM have been widely used to treat IPF, increase CER, and improve patients’ quality of life. However, there is a lack of comprehensive and systematic evidence to support the beneficial effects of CMI in combination with WM in IPF. 

In this study, NMA was used to systematically evaluate the efficacy of CMIs combined with WM in the treatment of IPF. Meanwhile, WM treatment of IPF mainly involves pharmacological treatment (e.g. glucocorticoids, anti-fibrotic drugs) and non-pharmacological treatment (e.g. oxygen and ventilators). Different outcome indicators such as CER, PaO2, PacO2, pulmonary function test (DLCO, FVC, TLC, FEV1%, and FEV1/FVC%), and molecular composition (IIIC and TGF)-associated fibrosis were analyzed to identify drugs with more efficacy advantages and provide a reference basis for the selection of CMIs for patients with IPF. A total of 63 RCTs involving 14 types of CMIs were eventually included in this NMA for data analysis. As shown in Table 3, the SUCRA of DH+WM in CER, PaO2, PaCO2, TLC, and TGF were 61.4 %, 83.7 %, 83.3 %, 89.0 %, and 85.7 %, respectively, ranking in the top 3. In addition, the SUCRA of SX+WM in CER, PaO2, PaCO2, FVC, FEV1%, and TGF were 75.5%, 71.7%, 99.8%, 86.2%, 85.9%, and 75.3%, ranking in the top 3. In terms of CER, SXN+WM, SXT+WM, SX+WM, DH+WM, and HQ+WM were the top five treatment strategies with the highest probability of being the best intervention. Furthermore, SXN combined with WM had the best performance in the CER (91.1%) and FVC (99.9%); HHS combined with WM had the best performance in the PaO2 (91.4%), FEV1/FVC% (93.0%), and IIIC (94.5%), all of which are important options for the treatment of IPF.

Reviewer: 2

#1. Authors must concentrate on the formatting, and use of symbols, etc., throughout the manuscript.

Response: Thank you so much for your comment. I had carefully checked the entire manuscript and made changes so that our manuscript meets PLOS ONE's style requirements which were marked in red in the paper. 

#2. Abstract should be revised and it should be more focus on your aim and results. Introduction and last paragraph of the introduction section should be elaborated in detail for better understanding of your studied issue.

Response: Thank you for your suggestion. I have revised the abstract to better reflect the aims and findings of this study. In addition, I have added some expressions to highlight our studied issue in the introduction section. Details as follows. 

1 Introduction

Idiopathic pulmonary fibrosis (IPF) is a chronic, progressive and lethal fibrotic lung disease, characterized by diffuse alveolitis, profound changes in epithelial cell phenotype and fibroblast proliferation. The incidence of IPF is around 8/10 million-15/10 million, accounting 65% interstitial lung disease[1]. IPF mostly presents as a chronic disease, but patients have an average median survival of only 2-4 years after diagnosis. Because of its unclear pathogenesis, the treatments for IPF are limited and causing high rate of mortality[2].

Glucocorticoids can relieve IPF patients' symptoms, but it is ineffective in reversing the lung damage. Lung transplantation is the last treatment for IPF patients. More effective therapeutic ways are becoming available for IPF patients following the research progress on pathogenesis of IPF.[3]. Pirfenidone and nidanib are approved for the treatment of IPF because they can slow down the decline of lung function and disease progression; however, these two drugs have more adverse effects, and no reliable evidence has been found to confirm that they significantly improve patients' symptoms and quality of life. Moreover, the cost of both drugs is high, which places a heavy economic burden on patients and society [4]. Traditional Chinese medicine (TCM) has a long history of treating IPF, and treatment is based on syndrome differentiation, which has the advantages of low toxicity, multi-level and multi-target, and unique advantages in clinical application [5,6].

Clinical trials of TCM in the treatment of IPF are gradually increasing, but it is unclear whether they can slow down the progression of the disease. Based on the research method of evidence-based medicine, this study used NMA to compare the number of different CMIs combined with WM interventions under the same conditions to obtain more reliable evidence for clinical reference. NMA is a development of the traditional meta-analysis, which has the advantage of providing quantitative statistical analysis of different interventions for the same disease and ranking them in order of efficacy, thus providing evidence to support the clinical use of drugs. Therefore, This study aims to provide evidence-based clinical practice by collecting RCTs on the efficacy and safety of CMIs combined with WM in the treatment of IPF.

#3. More abbreviation makes difficult to understand the concept. Please revise it.

Response: Thank you so much for your comment. I have revised the abbreviation to better understand the concept. Details as follows.

HHS can promote fibrin degradation and fibrotic damage within lung tissue, and delay the decline of lung function by reducing the levels of hyaluronic acid (HA), laminin (LN), and type Ⅲ procollagen (PCIII).

The flavonoids in Ginkgo biloba extract can significantly increase blood oxygen concentration, which has an inhibitory effect on serum interleukin- 6 (IL6), IL8, and tumor necrosis factor-α (TNF-α) levels in patients with IPF.

#4. Framework figure is required. It will be useful to the readers for better understanding of the studied issue.

Response: Thank you so much for your comment. We have added this research framework of the study, which is shown in Fig. 1A.

 #5. In conclusion section authors should provide some future prospectus related to present study.

Response: Thank you so much for your comment. We have revised the conclusion section. Details as follows.

5. Conclusions

IPF is a slowly progressive lung disease with a high mortality rate and an unknown pathogenesis. There is no effective cure for IPF, and CMIs combined with WM is expected to be one of the future strategies for IPF treatment. This NMA confirms that the 14 treatment strategies used in this study improved the symptoms and clinical indicators of patients with IPF in different ways. The combined use of DH, SXN, SX, and HHS with WM exerted a more positive effect in treating IPF than that done by WM alone. However, it is worth noting that since different types of CMIs have different effects and functions in patients, patients’ conditions should also be considered while selecting the CMIs to play a better role in the treatment of IPF. 

#6. Authors should clearly state the figure caption as well as description for ease of understanding of the readers.

Response: Thank you so much for your comment. We have revised the the figure caption. Details as follows. Further details were detailed in the full article.

Fig. 1A. The research framework of the study

Fig. 1B. Flow chart of the search for eligible studies.

Fig. 2. Network relationship diagram of each outcome index of CMIs combined with WM in the treatment of IPF. A, CER; B, PaO2; C, PaCO2; D, DLCO; E, FVC; F, TLC; G, FEV1%; H, FEV1/FVC%; I, IIIC; and J, TGF.

We tried our best to improve the manuscript and made some changes in the manuscript. We appreciate for your warm work earnestly, and hope that the correction will meet with approval. Once again, thank you very much for your comments and suggestions. 

Best regards,

Hongsheng Cui

---

## [Decision Letter · Decision Letter 1]

13 Jul 2022

Efficacy of traditional Chinese medicine injections for treating idiopathic pulmonary fibrosis : a systematic review and network meta-analysis

PONE-D-22-09987R1

Dear Dr. Cui,

We’re pleased to inform you that your manuscript has been judged scientifically suitable for publication and will be formally accepted for publication once it meets all outstanding technical requirements.

Kind regards,

Jen-Tsung Chen, Ph.D.

Academic Editor

PLOS ONE

Additional Editor Comments (optional):

Reviewers' comments:

Reviewer's Responses to Questions

**Comments to the Author**

1. If the authors have adequately addressed your comments raised in a previous round of review and you feel that this manuscript is now acceptable for publication, you may indicate that here to bypass the “Comments to the Author” section, enter your conflict of interest statement in the “Confidential to Editor” section, and submit your "Accept" recommendation.

Reviewer #1: (No Response)

Reviewer #2: All comments have been addressed

2. Is the manuscript technically sound, and do the data support the conclusions?

Reviewer #1: (No Response)

Reviewer #2: Yes

3. Has the statistical analysis been performed appropriately and rigorously? 

Reviewer #1: (No Response)

Reviewer #2: Yes

4. Have the authors made all data underlying the findings in their manuscript fully available?

Reviewer #1: (No Response)

Reviewer #2: Yes

5. Is the manuscript presented in an intelligible fashion and written in standard English?

Reviewer #1: (No Response)

Reviewer #2: Yes

6. Review Comments to the Author

Reviewer #1: (No Response)

Reviewer #2: Authors have addressed all my queries as well as they have suitably revised the manuscript. Therefore, I recommend this manuscript can be accepted for publication in its current form.

7. PLOS authors have the option to publish the peer review history of their article (what does this mean?). If published, this will include your full peer review and any attached files.

Reviewer #1: No

Reviewer #2: No

---

## [Editor Report · Acceptance letter]

15 Jul 2022

PONE-D-22-09987R1 

Efficacy of traditional Chinese medicine injections for treating idiopathic pulmonary fibrosis: a systematic review and network meta-analysis 

Dear Dr. Cui:

I'm pleased to inform you that your manuscript has been deemed suitable for publication in PLOS ONE. Congratulations! Your manuscript is now with our production department. 

Kind regards, 

on behalf of

Dr. Jen-Tsung Chen 

Academic Editor

PLOS ONE